# Advanced Cardiac Imaging and Women’s Chest Pain: A Question of Gender

**DOI:** 10.3390/diagnostics13152611

**Published:** 2023-08-07

**Authors:** Federica Dell’Aversana, Carlo Tedeschi, Rosita Comune, Luigi Gallo, Giovanni Ferrandino, Emilia Basco, Stefania Tamburrini, Giacomo Sica, Salvatore Masala, Mariano Scaglione, Carlo Liguori

**Affiliations:** 1Department of Precision Medicine, University of Campania “L. Vanvitelli”, 80138 Napoli, Italy; 2Operational Unit of Cardiology, Presidio Sanitario Intermedio Napoli Est, ASL-Napoli 1 Centro, 80144 Napoli, Italy; carlo.tedeschi@hotmail.it; 3Department of Radiology, Ospedale del Mare-ASL Napoli 1, 80147 Napoli, Italy; gianni.ferrandino90@gmail.com (G.F.);; 4Department of Radiology, Monaldi Hospital Azienda dei Colli, 80131 Napoli, Italy; 5Department of Medical, Surgical and Experimental Sciences, University of Sassari, 07100 Sassari, Italy; 6Department of Radiology, James Cook University Hospital, Middlesbrough TS4 3BW, UK

**Keywords:** cardiovascular diseases, ischemic heart disease, gender medicine, coronary computed tomography angiography, cardiac magnetic resonance imaging, coronary artery disease, myocardial infarction, spontaneous coronary artery dissection, myocardial infarction with non-obstructive coronary arteries, takotsubo syndrome

## Abstract

Awareness of gender differences in cardiovascular disease (CVD) has increased: both the different impact of traditional cardiovascular risk factors on women and the existence of sex-specific risk factors have been demonstrated. Therefore, it is essential to recognize typical aspects of ischemic heart disease (IHD) in women, who usually show a lower prevalence of obstructive coronary artery disease (CAD) as a cause of acute coronary syndrome (ACS). It is also important to know how to recognize pathologies that can cause acute chest pain with a higher incidence in women, such as spontaneous coronary artery dissection (SCAD) and myocardial infarction with non-obstructive coronary arteries (MINOCA). Coronary computed tomography angiography (CCTA) and cardiac magnetic resonance imaging (CMR) gained a pivotal role in the context of cardiac emergencies. Thus, the aim of our review is to investigate the most frequent scenarios in women with acute chest pain and how advanced cardiac imaging can help in the management and diagnosis of ACS.

## 1. Introduction

Cardiovascular diseases (CVDs) are the leading cause of death in both men and women, reaching 19.1 million deaths/year globally [1]. With 2.2 million deaths in women and 1.9 million in men, CVD remains the most common cause of death also in European countries; in women, ischemic heart disease (IHD) is the most common cause, accounting for 38% [2].

Although mortality in both sexes has gradually decreased, according to the literature, young women have a worse outcome because they are often undiagnosed. The biological variances among men and women, also called sex differences, are based on the combination of different hormonal and genetic expression and typically are responsible for the 10-year delay in onset of acute coronary syndrome (ACS) in women compared to men [3]. Another relevant sex discrepancy, according to the INTERHEART study, is the presence of a greater burden of CVD risk factors in women compared to men of the same age [4,5]. 

Several studies demonstrated that also traditional CVD risk factors such as diabetes, hypertension, dyslipidemia, smoking, obesity, overweight, and physical inactivity have a worse impact on female patients compared to men [6,7,8]. The Framingham Heart Study showed that obesity increases the relative risk of coronary artery disease (CAD) by 64% in women compared with 46% in men and that the atherosclerotic cardiovascular disease (ASCVD) risk factor status of diabetic women is more adverse than in diabetic men [9,10,11]. Conventional CVD risk factors impact women more adversely also because they are less often subjected to primary prevention and are typically treated less appropriately and intensively compared to men of the same age [3,12,13,14,15,16]. 

The risk of cardiac events is increased by emerging non-traditional female-specific CVD risk factors that often go underrecognized, such as preterm delivery, gestational diabetes, hypertensive pregnancy disorders, persistence of weight gain after pregnancy, depression, autoimmune diseases, radiation therapy, and chemotherapy [3,17]. 

The diagnosis of ACS can be especially difficult in young women since, in this group, the probability of CAD and acute myocardial infarction (AMI) is low, and the presence of atypical symptoms can lead to misinterpretation. Although acute chest pain is the most common symptom of AMI, there is a higher prevalence of atypical symptoms in women, such as pain in the neck and upper back, fatigue, shortness of breath, nausea, and vomiting, resulting frequently in a delayed diagnosis [18]. 

Usually, myocardial ischemia is identified with CAD, a traditional male pattern, which is considered a flow-limiting obstructive pathology relative to epicardial coronary arteries. Anyway, obstructive CAD is only a part of the wider range of conditions gathered under the definition of IHD, which typically affects women more frequently. IHD includes several non-obstructive CAD patterns, such as microvascular dysfunction (CMD), endothelial dysfunction, vasomotor anomalies, Spontaneous Coronary Artery Dissection (SCAD), and Takotsubo syndrome (TTS). As demonstrated by the ISCHEMIA trial, women usually have smaller vessels with higher vascular stiffness, fewer collateral branches, and lower coronary flow reserve. These anatomical characteristics partially explain the higher angina incidence in women who present with non-obstructive CAD [19].

Furthermore, the MESA study confirmed that, at the same age and risk factor level, women have lower coronary artery calcification (CAC) scores corresponding to high-risk plaques, as evidenced by CV mortality, which is 1.3 times greater than that present in a man with the same CAC [20,21]. Cardiac computed tomography angiography (CCTA), according to European Society of Cardiology (ESC) and American College of Cardiology (ACC) guidelines, has recently become the first-step evaluation technique in patients presenting acute chest pain, according to a low-to-intermediate pretest disease probability [22,23]. On the other hand, the American Heart Association (AHA) chest-pain guidelines stated the utility of cardiac magnetic resonance imaging (CMRI) in the evaluation of patients in specific scenarios [24]. 

Cardiac advanced imaging is nowadays a fundamental diagnostic instrument in the evaluation of patients showing a coronary or myocardial suspected issue more so in the female sex. In this review, we summarize the specific features of cardiac advanced imaging, intended as CT and MRI, in three of the most common scenarios related to acute chest pain in the female population: coronary heart disease, stress cardiomyopathy, and inflammatory myocardial and pericardial disease [25].

## 2. Acute Coronary Syndrome

Acute coronary syndrome (ACS) refers to a wide spectrum of clinical presentations, ranging from ST-elevation myocardial infarction (STEMI) to non-ST elevation myocardial infarction (NSTEMI) and unstable angina. It is almost always associated with the rupture of an atherosclerotic plaque and partial or complete thrombosis of the related artery [26].

Suspicion of ACS arises in patients who complain of acute chest discomfort and chest-pain equivalent symptoms such as pain in the left arm, epigastric pain, and dyspnea [22]. However, the clinical manifestations may differ between men and women: the latter are more likely to present with atypical symptoms, such as upper back and neck pain, weakness, nausea, and vomiting [18,27].

According to 4th Universal Definition of Myocardial Infarction (UDMI), the pathogenetic mechanism responsible for myocardial ischemia in myocardial infarction type 1 (MI1) is coronary artery disease (CAD) complicated by atherothrombotic plaque disruption in CAD. Myocardial infarction type 2 (MI2) is caused by the mismatch between oxygen supply and demand [28]. 

The detection of blood levels of cardiac troponin cTn above 99th percentile upper reference is proof of myocardial necrosis and allows us to make a diagnosis of acute MI in patients with symptoms of myocardial ischemia and supportive ECG changes [29]. Based on ECG findings, it is possible to classify MI as NSTEMI or STEMI to ensure an adequate therapeutic approach. Persistent ST elevation in patients with acute chest pain is usually due to total or subtotal coronary artery occlusion that will lead, in most patients, to STEMI; these patients should be reperfused immediately with primary percutaneous intervention (PCI) or fibrinolytic therapy. Acute chest discomfort in the absence of ST-segment elevation is defined as NSTEMI. A diagnosis of unstable angina is made in the presence of symptoms of myocardial ischemia without cell necrosis [30].

MI1 is caused by atherothrombotic plaque disruption, with erosion being more frequent than rupture in young women than in men, as evidenced in a secondary analysis from the PROMISE trial [31]. Typically, MI1 develops in patients with underlying CAD; however, another gender difference is that ACS, and especially MI1 in women, occurs predominantly in non-obstructive CAD defined as <50% stenosis of epicardial vessels [32]. 

The WISE study demonstrated the absence of obstructive CAD at ICA in most women (57%) with symptoms and signs of myocardial ischemia [33]. The lower incidence of obstructive CAD (11.5% vs. 29.8%) and the higher prevalence of normal coronary arteries (49.6% vs. 26.2%) were demonstrated in a post hoc analysis by the SCOT-HEART trial where participants with suspected CAD were studied using coronary computed tomography angiography (CCTA) [34]. 

CCTA has become a valid noninvasive method to detect, characterize, and quantify coronary atherosclerosis, allowing us to identify high-risk plaques (HRPs) that are statistically associated with a higher number of adverse events [35,36]. In addition, the ROMICAT II trial demonstrated that women who undergo CCTA had lower hospitalization rates with shorter hospital stays and a lower total radiation level than men [37]. Data from PARADIGM registry showed a lower non-calcified plaque progression, with a slower progression of CAD [38]. The lower incidence of obstructive CAD in women makes the diagnosis of ACS challenging. 

In females, rather than CAD, it might be more appropriate to use the ischemic heart disease (IHD) definition to include all the pathophysiology responsible for myocardial injury [39]. 

In type 2 MI, there is an imbalance between oxygen supply and demand that is often generated from a reduction of myocardial perfusion that can be caused by diffuse CAD, coronary artery spasm, coronary microvascular dysfunction, coronary artery dissection, and coronary embolism. The oxygen supply may also be reduced for systemic causes such as bradyarrhythmia, severe anemia, and hypotension and/or shock; in addition, myocardial oxygen demand may be increased for severe hypertension or sustained tachyarrhythmia [28]. Typically, type 2 MI is related to a poor outcome, and it is more common in women [40,41]. 

The diagnosis of type 2 MI is often made after employing the ICA technique, excluding a ruptured plaque with thrombus in the artery supplying the damaged myocardium, and often the pathogenetic mechanism is not diagnosed [28]. CCTA in this clinical scenario allows us to exclude a significant obstructive CAD when symptoms suggestive of myocardial ischemia are present (Figure 1).

The low specificity of CCTA in determining the functional significance of identified coronary stenosis has been overcome by the introduction of CT stress myocardial perfusion imaging or CT-guided fractional flow reserve (FFR) computed tomography angiography (Figure 2 and Figure 3) [42,43,44]. The ADVANCE study results highlighted the presence of gender differences in the functional significance of coronary stenosis since women have a higher CT-FFR with the same degree of stenosis than their male counterparts [43]. 

Hence, a paradigm shift is needed to bring about a new gender-specific interpretation of CAD [45]. The leading role in the evaluation of the functional impact of a coronary stenosis, among imaging techniques, is historically covered by the stress-CMR (S-CMR), even according to imaging guidelines [24]. S-CMR is a functional method which, after the induction of a pharmacological stress, allows us to identify any perfusion defects and wall motion abnormalities [46]. S-CMR has a sensitivity of 90% and specificity of 80% for the detection of anatomically significant CAD and a sensitivity of 89% and specificity of 87% to identify functionally significant CAD [47]. In cases of doubtful presentation, CMR can be performed after the event to assess the presence of jeopardized myocardium on T2-weighted sequences, which can be considered the expression of wall edema, a sign of acute myocardial damage [48,49]. CMR mainstream, in the case of suspected infarction, is represented by the possibility to characterize heart tissue thanks to late gadolinium enhancement (LGE) imaging, which, associated with the mapping techniques and the evaluation of the myocardial edema, allows us to make a differential diagnosis between takotsubo syndrome (TTS), myocardial infarction with non-obstructive coronary arteries (MINOCA), MI1, and MI2 [50]. CMR permits us to evaluate the regional wall motion abnormality (WMA) patterns, volumes, and functions of the ventricles, as well as complications such as thrombus formation and ventricular aneurysm; and help us to differentiate ACS from non-ACS related causes of chest pain [51,52].

## 3. Myocardial Infarction with Non-Obstructive Coronary Arteries (MINOCA)

Myocardial infarction with non-obstructive coronary arteries (MINOCA) is a subtype of infarction characterized by clinical and biochemical evidence of myocardial injury without significant visible obstructions on angiograms [28,53]. Approximately 6–15% of all myocardial infarctions are caused by MINOCA [54], with a higher incidence in specific groups, such as younger patients, women, and those without traditional risk factors for coronary artery disease (CAD) [55], as stated by the data collected in the VIRGO registry [56]. According to fourth universal definition of acute myocardial infarction (AMI), the term MINOCA is restricted to patients whose clinical presentation is due to an ischemic cause [28]; the diagnosis is achieved by excluding other causes of elevated cardiac troponin (such as sepsis, pulmonary embolism, aortic dissection, and cardiac contusion), undiagnosed obstructive CAD, and non-ischemic causes of myocardial injury [55]. 

MINOCA should be considered a working diagnosis after excluding significant CAD. Then, further tests should be performed to evaluate myocardial viability through late iodine or gadolinium enhancement or LV function by CMR, which allows for the underlying cause of MINOCA to be diagnosed in 87% of cases [22]. MINOCA can have various underlying causes, each contributing to inadequate blood flow to the heart muscle and resulting in myocardial injury [56]. These causes include coronary artery spasm, microvascular dysfunction, embolisms, inflammation, and plaque disruption without significant luminal stenosis. These factors collectively contribute to the development of MINOCA [57]. 

Hormonal factors, genetic predispositions, and autoimmune conditions have been proposed as potential contributors to MINOCA in women. Estrogens have cardioprotective effects, maintaining vascular health and promoting vasodilation: the fluctuations in their levels during different phases of a woman’s menstrual cycle and during menopause may contribute to coronary artery spasm or microvascular dysfunction, both of which represent common causes of MINOCA [58]. Women generally have smaller coronary arteries, making them more susceptible to endothelial dysfunction and microvascular abnormalities. These discrepancies may result in an impaired regulation of the coronary blood flow, increasing the risk of MINOCA [59]. Furthermore, women with MINOCA may often present atypical symptoms, such as fatigue, shortness of breath, and upper abdominal discomfort, posing challenges for diagnosis. These differences in symptom presentation can lead to an under recognition of MINOCA in women, as healthcare providers may be less likely to suspect a heart-related issue. Further research is needed to gain a more comprehensive understanding of all of these factors and to develop tailored diagnostic and treatment strategies for MINOCA, particularly in women [60]. 

Several studies supports the early use of CMR in MINOCA patients to find the underlying cause responsible for the myocardial ischemic event, with a diagnostic accuracy of about 90% when performed within a week from the index event [61,62]. Furthermore, another noninvasive imaging tool is coronary computed tomography angiography (CCTA); although it is not able to identify plaque rupture or erosion, it may be considered in the diagnostic workup to identify CAD after the ICA has excluded significant stenosis (>50%) (Figure 4) [57].

Hence, advanced cardiac imaging should be considered in the case of MINOCA in order to reach a faster diagnosis and to assess the underlying causes of disease, resulting in a prompt treatment and, finally, reducing patients’ morbidity.

## 4. Spontaneous Coronary Artery Dissection (SCAD)

SCAD is a non-iatrogenic, nontraumatic, and nonatherosclerotic dissection of the epicardial coronary artery wall resulting in hematoma formation within or between the intima, media, or adventitia [63]. There are two main hypotheses trying to explain the intramural hematoma (IMH) formation: the intimal tear hypothesis and the medial hemorrhage hypothesis. In both cases, the resulting false lumen can compress and/or obstruct the true lumen, leading to the occlusion of the coronary artery and resulting in myocardial ischemia and necrosis. Thrombosis of the true and/or false lumen may contribute to the pathophysiology of SCAD [64].

There are three types of SCAD according to the Saw angiographic SCAD classification. Type 1 refers to the classic appearance of multiple radiolucent lumens or arterial wall contrast staining, type 2 is characterized by the presence of diffuse stenosis varying in length and severity, and type 3 involves the presence of a tubular or focal stenosis that mimics atherosclerosis [63,64]. 

SCAD typically affects patients with few or no traditional cardiovascular risk factors, in whom suspicion for ACS is low and therefore often goes undiagnosed, as confirmed by a recent meta-analysis in which FMD was present in 68% of patients, and hypertension in 45% with a diagnose of SCAD [65]. The estimated prevalence of SCAD as the cause of myocardial infarction is approximately 0.1–0.4% and accounts for 0.4% of sudden cardiac death [66]. SCAD predominantly occurs in women, accounting for about 35% of MIs in women under the age of 50 and representing the most common cause of pregnancy-related Mis [63]. The incidence is higher, approximately 22–35% of ACS, in women younger than 60 years, especially in the peripartum period, in patients with a history of fibromuscular dysplasia (FMD), anxiety, depression, or psychiatric conditions. 

Pregnancy-associated SCAD (P-SCAD) represents 5% of all SCAD and has a higher incidence in multiparous women and women in the early postpartum period [22,66,67]. Given the subpopulation affected by SCAD, characterized by a low incidence of traditional cardiovascular risk factors, the etiology of SCAD is multifactorial. Hereditary or acquired arteriopathies, systemic inflammatory diseases, genetic factors, and hormonal influences seem to be predisposing factors that weaken the walls of the arteries on which external precipitating factors can then act [63,68]. The most frequent coexisting condition, reported in up to 86% of SCAD cases, is FMD [64]. Left main and left anterior descending are the coronary arteries most often involved in women with P-SCAD. Compared with those with non-pregnancy-related SCAD, P-SCAD patients present more proximal arterial dissections, more extensive infarcts, lower mean left ventricular ejection fraction, and an overall poorer prognosis [69,70]. In a Mayo Clinic study, 54 women with P-SCAD were compared with 269 non-pregnancy-related SCAD women, with the former being younger, with left main and multivessel dissections, much more likely to have STEMI, worse LV function also at follow-up, and a lower incidence of FMD [70].

Clinical presentation depends on dissection extension and on the entity of flow compromission; the most common symptoms are chest discomfort and the elevation of troponin levels after emotional or physical stress or during pregnancy [22]. Invasive coronary angiography (ICA) is a widely available technique and is recommended for the early invasive management of ACS. During an angiographic exam, in a patient with SCAD, it is possible to see multiple radiolucent lumens and extraluminal contrast staining; these findings of SCAD enter the differential diagnosis with normal coronary arteries, atherosclerotic disease, coronary vasospasm, takotsubo syndrome, and thromboembolism [63,64]. 

The diagnosis of ACS due to SCAD compared to ACS secondary to atherosclerotic disease is fundamental given the different management both in the short and long term. Although the optimal management of SCAD is still unclear, early diagnosis is crucial, and a conservative approach should be preferred in non-high-risk patients. In fact, technical success rates of patients diagnosed with acute SCAD undergoing PCI are reduced compared to cases of atherosclerotic ACS (62% vs. 92%), and the rate of spontaneous vascular recovery means that patients with SCAD and preserved distal flow can be treated conservatively [3,71,72]. 

In a 2018 consensus document, the ACC validated the potential role of CCTA in the study of coronary arteries in patients with stable and recurrent forms of SCAD. The ECG triggered CCTA [66]. Some studies have validated the role of CCTA in patients with SCAD; for example, Sun et al. demonstrated that CCTA was found to be more diagnostic than ICA in SCAD, while Roura et al. validated the role of CCTA in follow-up 3–6 months after SCAD [63,73,74]. CCTA findings in an SCA scenario are classified into primary and secondary (Figure 5, Figure 6 and Figure 7). The primary ones include abrupt luminal stenosis present in 64% of patients and tapered luminal stenosis present in 36%, IMH in 50%, and dissection flap in 14%. Secondary CCTA findings that increase the reader’s confidence for the diagnosis of SCAD include epicardial and perivascular fat stranding, myocardial hypoperfusion, coronary tortuosity, the absence of coronary calcifications, and an ill-defined area of hyperattenuation adjacent to the artery on unenhanced CT acquisition [75,76]. 

The prognosis of patients with SCAD is substantially good: most patients are treated conservatively, and of these, only between 2.6% and 8.5% undergo revascularization during hospitalization. Primary revascularization is the primary treatment in 16.7% to 47.1% of patients, with success rates lower than those for patients with atherosclerotic ACS [77,78,79]. Alfonso et al. studied 45 patients over a 6-year period, demonstrating that the natural history of SCAD predicts spontaneous healing, making conservative treatment an adequate strategy [72]. The incidence of recurrent SCAD, considered to be subsequent de novo dissections in previously unaffected arteries, is approximately 11% of patients according to a cohort of patients studied in Vancouver, while the 10-year Kaplan–Meier estimated risk of recurrence is 29.4% [72,80]. 

MRI is a noninvasive imaging tool that can be helpful in diagnosing SCAD. Excluding a small proportion of patients with an angiographically confirmed SCAD diagnosis not showing signs of infarction upon MRI examination, the presence of delayed gadolinium enhancement in the distribution territory of the dissecting vessel can confirm the diagnosis [81]. The MRI coronary findings that are characteristic of SCAD are the presence of transmural, subendocardial, myocardial, patchy LGE; the presence of IMH; and microvascular obstruction [82,83].

## 5. Takotsubo Syndrome (TTS)

Stress cardiomyopathy, or takotsubo syndrome (TTS), is a reversible cardiomyopathy that is characterized by a transient left ventricle (LV) wall motion abnormality (WMA), with ischemic ECG changes in the absence of significant coronary artery disease (CAD) [84]. The name is derived from a Japanese octopus trap whose shape resembles the one assumed in systole by the LV during the TTS acute phase, with a narrow neck and apical ballooning [85]. The typical regional WMA pattern seen in TTS is linked to the presence of akinesia or dyskinesia of the mid-apical segments of the left ventricle walls, while basal segments are usually spared from motility alterations [86]. 

There are also less common patterns: (i) mid-ventricular ballooning pattern, (ii) basal ballooning pattern, (iii) biventricular pattern (iv), and focal ballooning pattern [87]. In most cases, TTS diagnosis, often clinically indistinguishable from acute coronary syndrome (ACS), is considered only after excluding other causes, and it is usually performed following the revised Mayo clinic criteria, which include (i) LV WMA and the presence of emotional or physical triggers; (ii) the absence of obstructive lesions of the coronary vessels; (iii) ECG changes with ST elevation and/or T-wave inversion; (iv) the elevation of cardiac markers; and (v) the absence of pheochromocytoma and myocarditis [88]. 

TTS is associated with myocyte damage, which is evidenced by the presence of an increase in the serum levels of cardiac biomarkers in the absence of significant CAD. For this reason, according to the 4th Universal Criteria of Myocardial Infarction, TTS has been classified as myocardial infarction with non-obstructive coronary arteries (MINOCA) [89]. An international consensus of experts has recently introduced a newer clinical score to help differentiate takotsubo syndrome (TTS) from acute coronary syndrome (ACS): the InterTAK Diagnostic Score. According to these criteria, patients typically show LV regional WMA dysfunction that extends beyond a single epicardial vascular distribution but can also be focal. The right ventricle may be involved, and new ECG abnormalities, such as ST-segment elevation, ST-segment depression, T-wave inversion, and QTc prolongation, are present, such as moderate and significant elevation of cardiac biomarkers (troponin and CK) and significant BNP levels [88]. 

Usually, the acute episode is precipitated by an emotional, combined, or physical trigger event, with the latter including many acute neurological pathologies and pheochromocytoma, while infective myocarditis should be excluded. Postmenopausal women are predominantly affected, and significant CAD does not necessarily rule out TTS [86,90].

Although the exact pathophysiological mechanism remains unclear, it has been supposed that acute myocardial stunning may be induced by the activation of the sympathetic nervous system, with a consequential release of catecholamines, inducing vasospasm of the epicardial coronaries, microcirculatory vessels, and direct myocardial toxicity, following an emotionally or physically stressful event [84,91,92,93]. In total, 80–90% of patients diagnosed with TTS are postmenopausal women [50]. This marked gender difference can be partly explained by the role that estrogens play in the endothelium-dependent and independent regulation of adrenergic tone and microvascular blood flow [50,84,93]. Even if TTS typically occurs in women older than 50, several cases have also been reported in younger women, especially in the peripartum period, most frequently after a cesarean section [51,94,95,96]. The incidence is also increasing due to the greater awareness of this syndrome: TTS is diagnosed in about 2–3% of all patients and in 5–6% of female patients with ACS symptoms and accounts for 0.5–0.9% of STEMI [50,97,98]. 

A diagnosis is reached after excluding the presence of a significant coronary artery obstruction upon the ICA, or showing that CAD is not capable of justifying WMA, as occurs in 15% of cases [28,85]. However, the assessment of the coronary arteries with ICA is mandatory to rule out CAD in patients with acute chest pain and acute changes upon an electrocardiogram [30,51,86]. CCTA has been proposed as an effective noninvasive alternative to ICA in stable patients with a low likelihood of ACS and in all clinical scenarios in which the suspicion of TTS is particularly high (recurrence suspected or acute critical illnesses), presenting ECG changes and elevation of cardiac biomarkers [86,99].

CCTA is a widely available noninvasive method that is characterized by a rapid execution, which allows for a comprehensive assessment of coronary artery anatomy. Moreover, cardiac CT allows for a multiplanar functional study of LV function and WMA distribution, overcoming the intrinsic limits of transthoracic echocardiography (TTE) and leading to evaluate myocardial late iodine enhancement (LIE) to characterize myocardial tissue and to identify LV thrombi [100,101]. The noninvasiveness of CCTA facilitates the study of fragile subjects who could not be subjected to ICA due to the high risk of complications [102], being also a valid aid to rule out other diseases presenting with acute chest pain, such as aortic dissection and pulmonary embolism [93]. 

In the acute phase of TTS, CMR is decisive in the differential diagnosis, with other causes of MINOCA requiring different treatments, such as myocarditis. A CMR study is mandatory in the post-acute phase in the case of persistence of WMA upon echocardiography, and/or, in the case of persistent changes, an ECG is used to confirm the diagnosis of TTS [30]. CMR allows us to assess myocardial contractile function on the cine sequences. 

The most common pattern of contractile dysfunction in TTS involves LV circumferentially extending beyond the distribution of a single coronary artery with a mid-ventricular-to-apical akinesia, hypokinesia or dyskinesia, and basal hyperkinesia [103]. Furthermore, a CMR study allows us to assess a possible right ventricle (RV) involvement, which is difficult to study at the TTE because of its complex geometry [87], the regional WMA patterns that typically affects patients with TTS, as well as to evaluate the volumes and functions of the ventricles, overcoming the intrinsic limits of the TTE (Figure 8). Therefore, CMR may represent a valid tool for both the diagnosis and follow-up of the evolution of TTS [51]. Indeed, with CMR, it is possible to highlight both the typical apical ballooning and the focal forms of WMA, which may be undiagnosed during an ultrasound examination [87]. 

Furthermore, a CMR study allows us to show if there is an involvement of the right ventricle (RV), which, due to its complex geometry, can also be missed during an ultrasound examination [87]. 

Secondary to the release of catecholamines and BNP, myocardial edema is typically found in the acute phase of TTS, representing a diagnostic parameter that allows us to evaluate the extension and severity of myocardial stunning. Myocardial edema gradually disappears with the recovery of normal myocardial contractility, and it is typically distributed in akinetic or dyskinetic segments of the myocardium, showing a transmural or diffuse distribution [104]. Edema and reversible inflammation of the myocardium can be identified as a signal hyperintensity on T2-weighted sequences involving the region with regional WMA that gradually normalizes over the course of a few weeks [87]. 

T2 mapping acquisitions can aid or serve as a substitute for classical inversion recovery acquisitions, contributing higher sensitivity as well [51,86,105]. Myocardial edema is present not only in TTS but also in acute myocardial infarction (AMI), albeit limited to a coronary territory, and in myocarditis, where it has a typical subepicardial distribution. A further difference for AMI and myocarditis is the absence of LGE and, therefore, of myocardial necrosis in the territories affected by RWMA in TTS [106,107].

## 6. Myocarditis

Myocarditis is an inflammatory condition affecting the myocardium, the muscular tissue of the heart [108]. Although previously considered a rare disease, nowadays the prevalence of myocarditis has been reported in 4–14 persons/100,000/years, with a mortality rate of 1–7% [109,110]. Myocarditis can be triggered by several factors, both infectious and non-infectious [111]. Its pathogenesis involves an inflammatory response within the myocardium that can lead to myocardial damage and dysfunction [108]. 

While myocarditis can affect individuals of all ages and both sexes, there is a higher prevalence of myocarditis in younger women [109]. This prevalence pattern may be associated with hormonal and immune system fluctuations that occur during different stages of a woman’s life. Hormonal changes, such as those occurring during the menstrual cycle, pregnancy, and menopause, have been proposed as potential contributors to this higher incidence [112]. Particularly, estrogens have been supposed to modulate immune responses and potentially influence the development and severity of myocarditis in women. Regarding immunological aspects, women generally exhibit stronger immune responses compared to men, potentially making them more susceptible to inflammatory processes. Variations in immune cell function, cytokine production, and immune signaling pathways could contribute to the increased incidence of myocarditis in women [113]. Other potential factors contributing to the higher incidence of myocarditis in women include genetic predisposition and differences in healthcare-seeking behavior [114]. 

Diagnosing myocarditis can be challenging due to its nonspecific symptoms and potential overlap with other cardiac conditions. However, advancements in imaging techniques have significantly improved diagnostic accuracy. In fact, myocarditis is the final diagnosis in about 35% of patients with an initial working diagnosis of MINOCA in which the finding of a non-ischemic pattern on CMR allows us to consolidate the diagnostic suspicion [115,116,117]. 

CMR is a useful tool to evaluate the presence and the extent of myocardial inflammation, assess ventricular function, and make a tissue characterization that can help to determine the extent of myocardial damage [116,117,118]. The role of CMR in noninvasive diagnosis of AM has been consolidated by the revised 2018 Lake Louise Criteria (LLC), which consist of major and supportive parameters. The major criteria are the presence of edema in T2-weighted sequences or T2 parametric mapping, early gadolinium enhancement, and fibrosis at LGE. Global or regional WMA, abnormal T1 and T2, and/or LGE and pericardial effusion are considered to be the supportive criteria of AM [116,119]. Typically, the inflamed myocardium has higher T1, T2, and ECV values, and the recently introduced CMR mapping techniques allow us to efficiently measure the relaxation time, both T1 and T2, allowing for a qualitative and quantitative assessment of myocardial inflammation. Furthermore, it is possible to estimate the extracellular volume (ECV) with the T1 maps acquired before and after the administration of a contrast agent adjusted for the hematocrit value [48,120,121,122,123]. 

A recent meta-analysis compared the diagnostic accuracy of LLC criteria in diagnosing acute myocarditis with native T1, T2, and ECV mapping, which appears to be comparable. The only parameter showing greater sensitivity than LLC is the prolonged T1 mapping [124]. This result was confirmed in a recent study: a prolonged T1 mapping showed a sensitivity of 82%, while pericardial effusion is the finding presenting the highest specificity (99%) [125]. Although the role of CMR in the diagnosis of myocarditis is consolidated, ECG-gated MDCT can be a valuable tool, especially in infarct-like presentations (Figure 9 and Figure 10), where it may be necessary to exclude CAD [126,127]. Using study protocols that include acquisitions 6-8 min after the administration of iodinated contrast medium, it is possible to evaluate the late iodine enhancement (LIE): the presence of focal or multifocal patterns of late myocardial hyperenhancement indicates myocardial damage, supporting the diagnosis of myocarditis (Figure 10) [128].

## 7. Pericarditis

Pericarditis is a condition that is characterized by inflammation of the pericardium, the thin sac that surrounds and protects the heart [129]. It can manifest as acute, subacute, recurrent, or chronic and is often associated with symptoms such as chest pain, shortness of breath, and palpitations [130]. Acute pericarditis accounts for most cases, while recurrent and chronic forms are less common [131]. The parietal layer of the pericardium is highly innervated, which means that any inflammatory condition caused by an infectious, autoimmune, or traumatic event can lead to intense retrosternal chest pain, a symptom frequently observed in cases of acute pericarditis [130]. 

According to the 2015 ESC guidelines for the diagnosis and management of pericardial diseases, acute pericarditis can be categorized into two primary groups based on its etiology: infectious causes and non-infectious causes [132]. Pericarditis is a relatively common condition, with an estimated annual incidence ranging from 10 to 30 cases per 100,000 individuals [133]. The incidence may vary among different populations and regions. Several studies have suggested a slightly higher incidence of pericarditis in men compared to women, but the condition can affect individuals of all ages and both sexes [129]. The role of gender in the epidemiology of pericarditis remains a subject of interest. While some studies have reported a higher incidence in men [129], others have suggested a female predominance [129,132,134]. Although the sex gap in cardiovascular diseases is a well-known phenomenon, the differences in incidence and severity of pericarditis between women and men are still debated. Guillen et al. reported the results of a retrospective study, aiming to evaluate the presence of specific sex and gender factors of pericarditis in women. Particularly, the authors enrolled patients diagnosed with acute pericarditis in an emergency setting. Interestingly, they reported that women presented a delayed time before first medical attendance; hence, they presented less frequently the typical electrocardiogram alterations of acute pericarditis. Moreover, women reported more frequent autoimmune pathogenesis than men [134]. Hormonal influences and genetic predisposition may contribute to these discrepancies. Further research is needed to gain a comprehensive understanding of the epidemiology of pericarditis, particularly its gender-specific aspects. 

Pericarditis diagnosis involves assessing clinical symptoms, performing a physical examination, and conducting diagnostic tests [132]. These tests may include an electrocardiogram (ECG), which often shows specific changes suggestive of pericarditis, such as ST segment elevation and PR segment depression. Blood tests, including inflammatory markers and cardiac enzymes, can help evaluate the underlying cause [135]. Imaging techniques such as echocardiography and cardiac magnetic resonance (CMR) aid in visualizing pericardial inflammation, thickening, and the presence of pericardial effusion [136]. CMR has emerged as a well-established diagnostic modality, providing valuable insights into pericarditis. It can detect pericardial thickening; edema/inflammation, using Short-TI Inversion Recovery-T2 weighted (STIR-T2w) imaging; and edema/inflammation or fibrosis through late gadolinium enhancement (LGE). Additionally, CMR can assess the presence of pericardial effusion, making it a comprehensive tool for objective pericarditis detection [137].

## 8. Cardiac Hybrid Imaging

Nuclear cardiac techniques such as single-photon emission computed tomography (SPECT) and positron emission tomography (PET) are routinely used in clinical practice to evaluate myocardial perfusion, especially in the asymptomatic, and the intermediate likelihood of CAD patients [138]. In fact, the role of both PET and SPECT to diagnose ischemic coronary artery disease is widely validated. SPECT can assess myocardial viability and accurately identify areas of abnormal perfusion [139,140]. On the other hand, myocardial perfusion imaging with PET permits us to evaluate regional and global myocardial perfusion, myocardial blood flow, and function at stress and rest [141]. The introduction of hybrid modalities such as SPECT/CT, PET/CT, and PET/MRI represents an attempt to overcome the limitations of single techniques by combining their strengths [142]. In fact, an ideal noninvasive technique to diagnose CAD should provide information on coronary anatomy and the hemodynamic severity of the lesion [141,142]. Besides serving as a functional evaluation based on the simultaneous assessment of cardiac function and perfusion, in hybrid devices, the addition of CT represents a useful tool. However, SPECT/CT and PET/CT are rarely employed in emergency scenarios, because of the logistic issues and low availability of scanners [142]. PET/MRI exploits the ability of MRI to characterize tissues in detail due to the high contrast of soft tissues and allows us to evaluate the functional parameters of the ventricles. If PET/MRI presents a relevant role in inflammatory focus imaging for sarcoidosis and myocarditis, still, few studies are performed in an IHD scenario [143]. Despite their enormous potential, hybrid techniques are still less available, especially in ED, and, in particular, PET/MRI research has been limited to feasibility data [144]. Further efforts should be made to standardize the role of SPECT/CT, PET/CT, and PET/MRI in the evaluation of acute chest pain, especially in women, with the aim of making it a single-stop exam in the study of the heart [145].

## 9. Future Perspectives 

Since its introduction, the role of dual-energy CT (DECT) in evaluating myocardial blood supply and making coronary arteries assessments has gradually increased [146]. DECT consists of the acquisition of two different sets of data using different energy levels. Material decomposition allows us to analyze tissue composition, resulting in a better distinction between iodine and other tissues [147]. DECT improves the characterization of plaque compared to conventional single-energy CT (SECT) by providing more detailed information on plaque composition. In fact, DECT can assess both the density and the atomic number of plaque constituents, thus representing a promising useful tool to detect vulnerable plaques in emergency scenarios, too [148]. If, on the one hand, DECT can be used to evaluate the morphology of a plaque, on the other hand, it can be used to assess the stenosis percentage [149]. DECT also allows for iodine quantification within myocardium since each human tissue and iodine-based contrast medium presents a typical absorption characteristic when penetrated with different X-ray energy levels [150]. Thus, DECT, used during the course of a perfusion study, has the capability to assess the hemodynamic significance of stenosis since it can gauge the iodine concentration in myocardium distally to stenosis [148]. In fact, DECT is the only imaging exam that permits us to assess ECV and coronary morphology simultaneously (Figure 11) [151]. The quantification of ECV in DECT is performed on iodine maps, with very accurate results comparable to those obtained at CMR (Figure 12) [152]. The role of DECT in non-emergency cardiac imaging is well established, allowing for a comprehensive analysis of coronary arteries’ morphology and changes in myocardial perfusion. The distribution of these CT scanners in the emergency department is still partly limited because of the higher cost and the need for dedicated software. However, the introduction of DECT into daily practice will offer a new reliable tool in cardiac emergencies [149]. DECT scanners should be recommended as an option for the evaluation of acute chest pain in women since they provide improved diagnostic accuracy and image quality, and, at the same time, they reduce both contrast media administration and radiation [148].

Photon-counting CT (PDC-CT) is a new CT-technology in which energy-integrating detectors (EIDs) used in current CT scanners have been replaced with energy-resolving detectors. Photon-counting detectors (PCDs) consist of a single thick layer of a semiconductor diode which directly generates electronic signals from incident X-ray photons. It must be considered that an increase in the CT matrix of up to 2048 × 2048 is possible using photon-counting scanners, reaching a significant improvement in spatial resolution of CT images [153,154]. PDC-CT overcomes many of limitations of conventional EIDs since it results in increased spatial resolution, which could help rule out CAD in the small vessels in order to detect tiny calcifications and high-risk features. Furthermore, PCD-CT can reduce common image artifacts, such as beam hardening and calcium blooming, facilitating the evaluation of severe coronary artery calcifications, without a radiation-dose increase [153,155]. In conclusion, PDC-CT seems to be a promising new technique which can undoubtedly be applied in the clinical scenario of acute chest pain in women.

## 10. Conclusions

The female sex is more prone, because of several risk factors mainly driven by hormonal agents, to specific pathologic events such as SCAD and TTS cardiomyopathy, which can be considered gender-related conditions. The etiopathogenesis of myocardial ischemia in women is more complex and heterogeneous than in men [25,156]. Often, women have a poorer prognosis because of two factors: the symptoms can be misunderstood, and there is the wrong popular idea that ACS is not common in women, even though CVD and, in particular, CAD are the main cause of morbidity and mortality in women. These factors combined can lead to mismanagement, underdiagnosis, and suboptimal treatment. In addition, women are less likely to be treated with an invasive coronary angiography (ICA); they are rarely revascularized and treated with adequate medical therapies [18,27]. The use of multi-modal imaging in the female sex must be encouraged because the use of combination data coming from ICA, CCTA, and CMRI makes it easier to reach a faster and precise diagnosis, allowing for correct patient management (Table 1). The combination of imaging information allows for the optimal interpretation of the underlying pathophysiologic process.

## Figures and Tables

**Figure 1 diagnostics-13-02611-f001:**
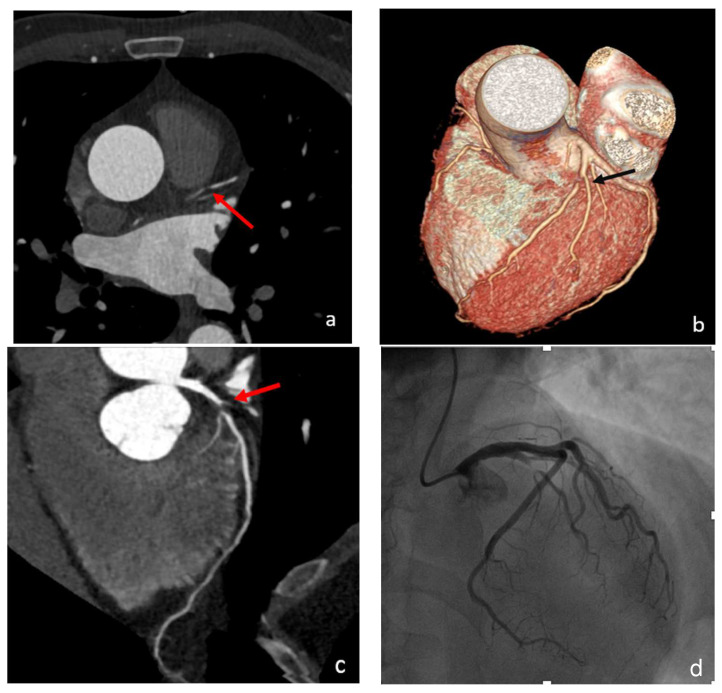
Case 1. ACS: A 37-year-old woman underwent a CCTA due to chest pain, ECG abnormalities, and elevated troponin levels. Imaging results, including axial two-dimensional imaging (**a**), 3D imaging (**b**), and CPR (curved planar reconstruction) (**c**), revealed the presence of a significant subocclusive stenosis (>90%) in the proximal LAD. This stenosis was caused by a non-calcified plaque (black and red arrows). A subsequent ICA (**d**) confirmed a significant stenosis with a subsequent vessel stenting.

**Figure 2 diagnostics-13-02611-f002:**
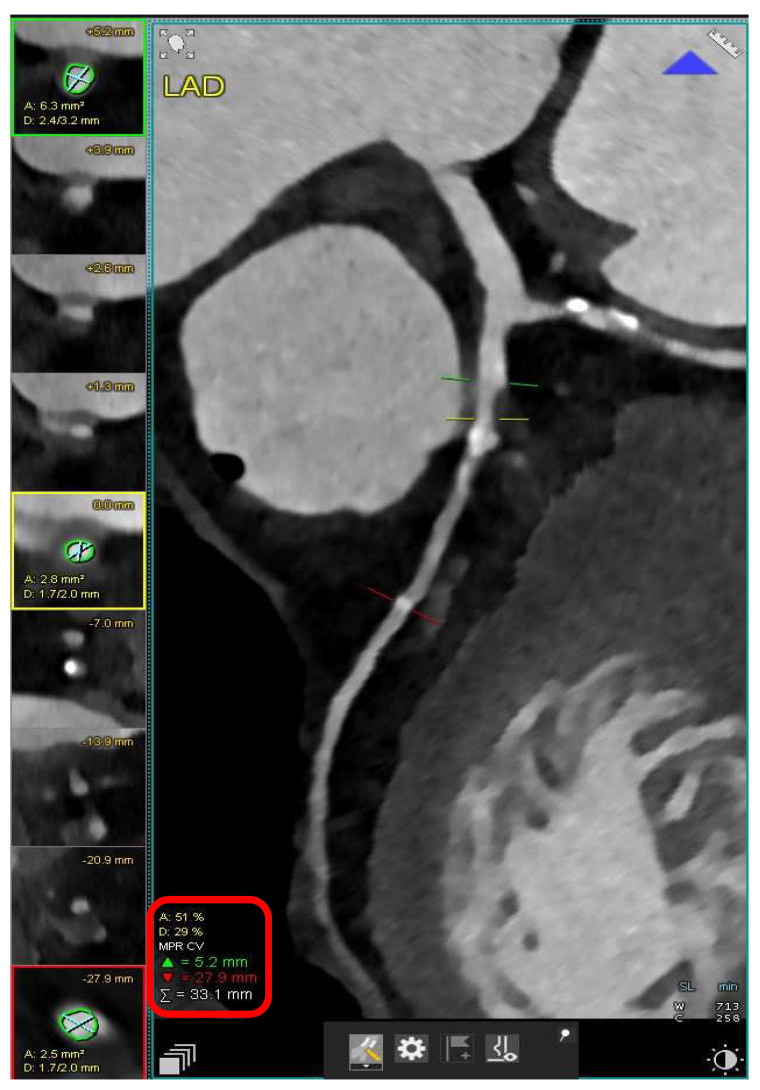
Case 2. Coronary computed tomography angiography (CCTA) was performed on a 73 years old woman who experienced acute chest pain. CT study revealed the presence of a diffuse, partly calcified atherosclerosis in LAD and a main stenosis in the proximal segment of the vessel, causing an intermediate morphological reduction of the patent lumen, accounting for 50–70% (stenosis area), highlighted in the red square.

**Figure 3 diagnostics-13-02611-f003:**
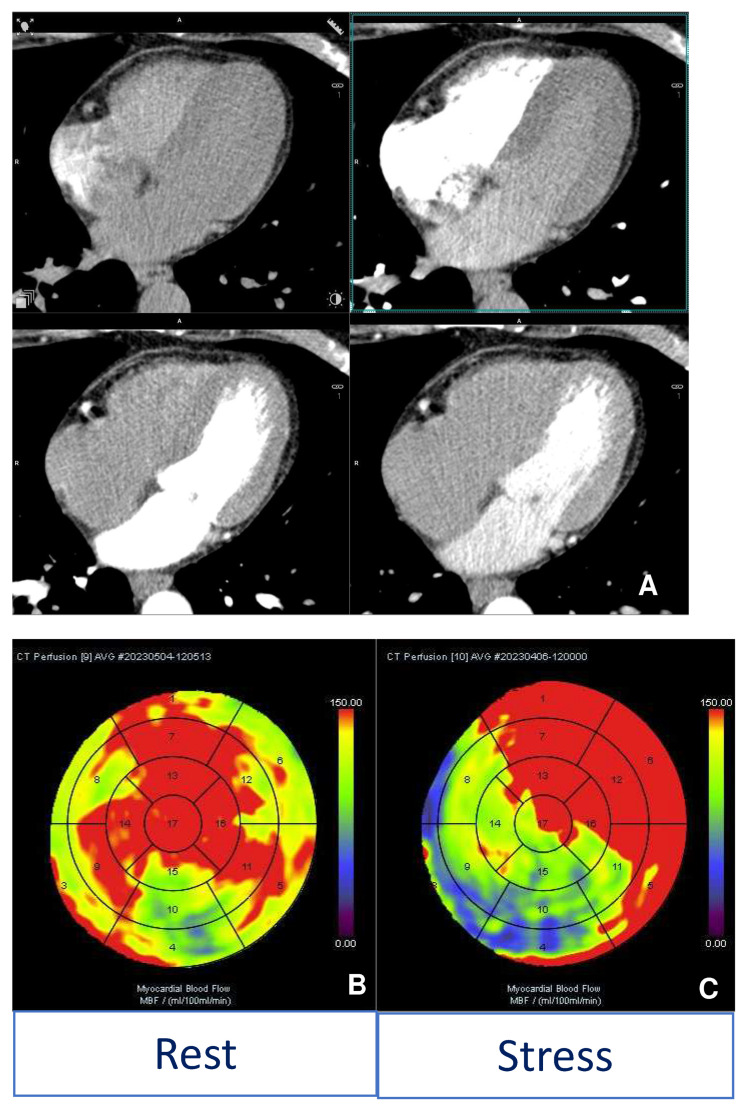
Case 2. To establish the real hemodynamic impact of the stenosis, a stress CT perfusion scan (**A**) was subsequently performed. The scan was composed by rest (**B**) and stress (**C**) dynamic acquisitions, using vasodilator coronary selective medication. A relative perfusion defect was seen in the stress bull eye (**C**) regarding interventricular septum, from the base to apex. The finding was clearly appreciable in comparison to the rest of the dynamic acquisition (**B**) and baseline angiographic reconstruction of the ventricle (**D**).

**Figure 4 diagnostics-13-02611-f004:**
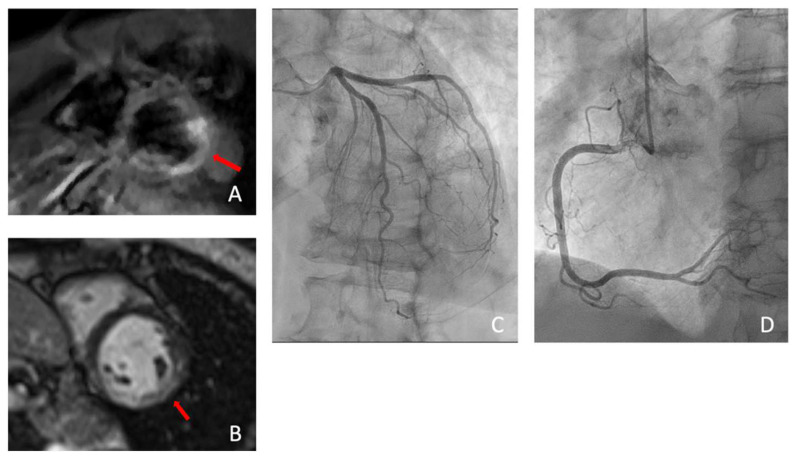
Case 3. MINOCA: A 55-year-old woman was admitted to the emergency department with chest pain, ECG changes, and mildly elevated necrosis enzymes. The patient underwent ICA, which did not show significant stenosis in the left (**C**) and right (**D**) coronary arteries. Given the persistence of symptoms, she was subjected to a CMR study, which showed acute myocardial injury at edema-sensitive sequences in the lateral wall of the left ventricle (red arrow (**A**)) and the presence of LGE (red arrow) on the same side (**B**), thus confirming MINOCA diagnosis.

**Figure 5 diagnostics-13-02611-f005:**
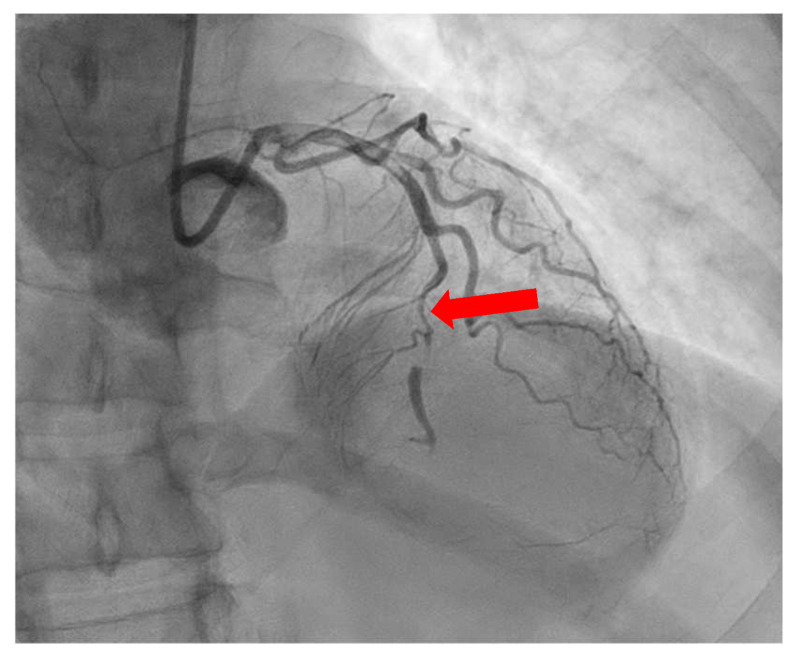
Case 4. SCAD: An invasive coronary angiography was conducted on a 56-year-old woman experiencing chest pain, elevated troponin levels, and ECG alterations. The figure depicts the presence of a segmental luminal interruption (red arrow) in the distal left anterior descending (LAD) artery with slow flow progression distally, representative of spontaneous LAD intimal dissection (Type 2).

**Figure 6 diagnostics-13-02611-f006:**
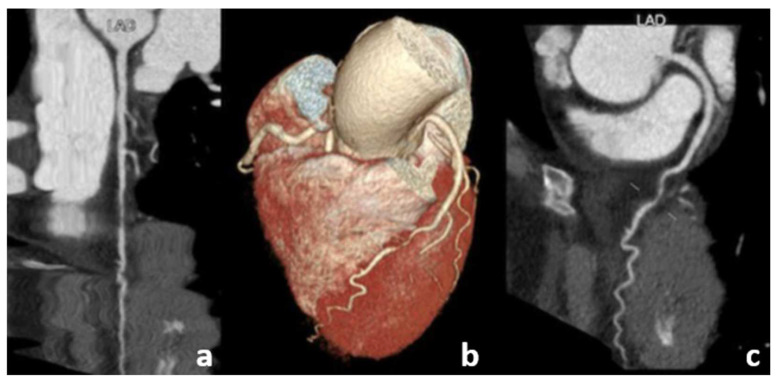
Case 4. SCAD: A CCTA follow-up was performed after 14 days. Curved planar reconstruction images (**a**,**c**), as well as a 3D reconstruction image (**b**), demonstrated a partial vessel reperfusion with moderate expansion of the luminal caliber.

**Figure 7 diagnostics-13-02611-f007:**
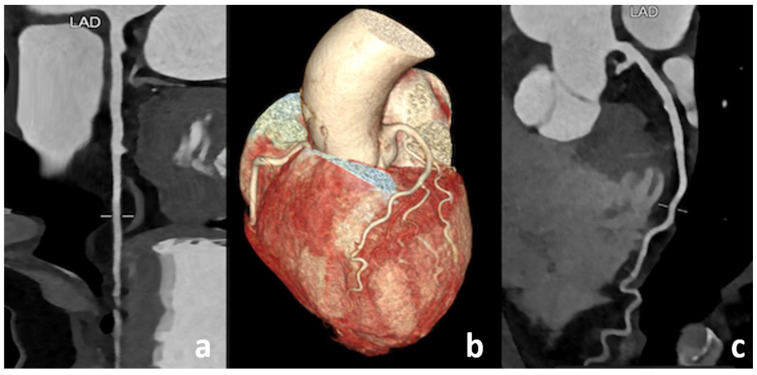
Case 4. SCAD: Another CCTA follow-up was performed 6 months later. Curved planar reconstruction images (**a**,**c**), as well as a 3D image (**b**), revealed the complete resolution of the dissection following conservative treatment.

**Figure 8 diagnostics-13-02611-f008:**
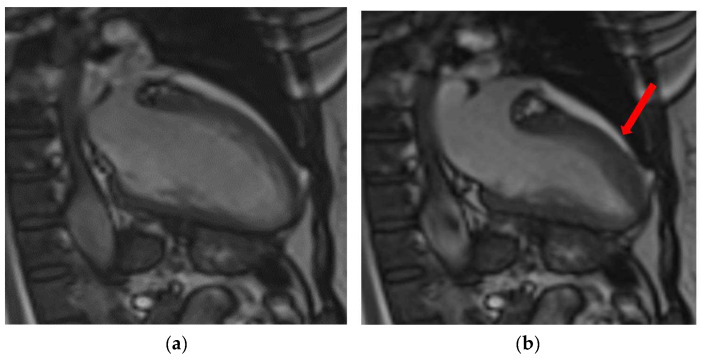
Case 5. Takotsubo Syndrome: Cardiac magnetic resonance (CMR) findings in a 56-year-old woman presented with chest pain and absence of coronary stenosis at ICA. In the end-systolic (**b**) phase, compared to end-diastolic (**a**) 2-chamber frames, there is noticeable mid-ventricular ballooning of the left ventricle (red arrow). This abnormality is accompanied by transmural wall edema (red arrow), demonstrated by the STIR sequence (**d**) in the absence of late gadolinium enhancement (LGE) (**c**).

**Figure 9 diagnostics-13-02611-f009:**
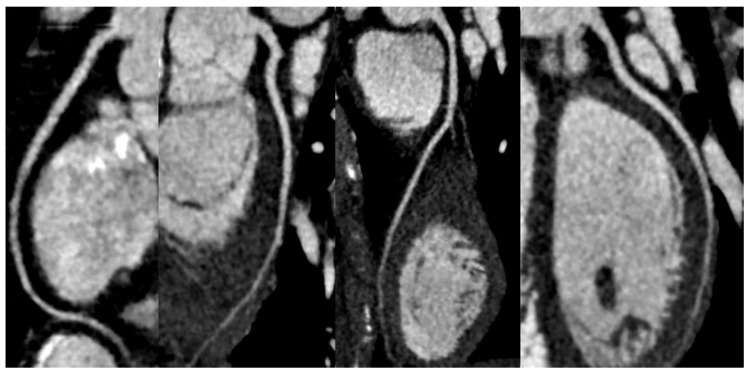
Case 6. Myocarditis: Coronary computed tomography angiography (CCTA) was performed on a 25-year-old woman who experienced a cardiac arrest, followed by cardiac reanimation, at the beach. The patient arrived at the hospital showing nonspecific ECG changes and a slight elevation in troponin levels. CT scan revealed normal patency and anatomy of the coronary tree.

**Figure 10 diagnostics-13-02611-f010:**
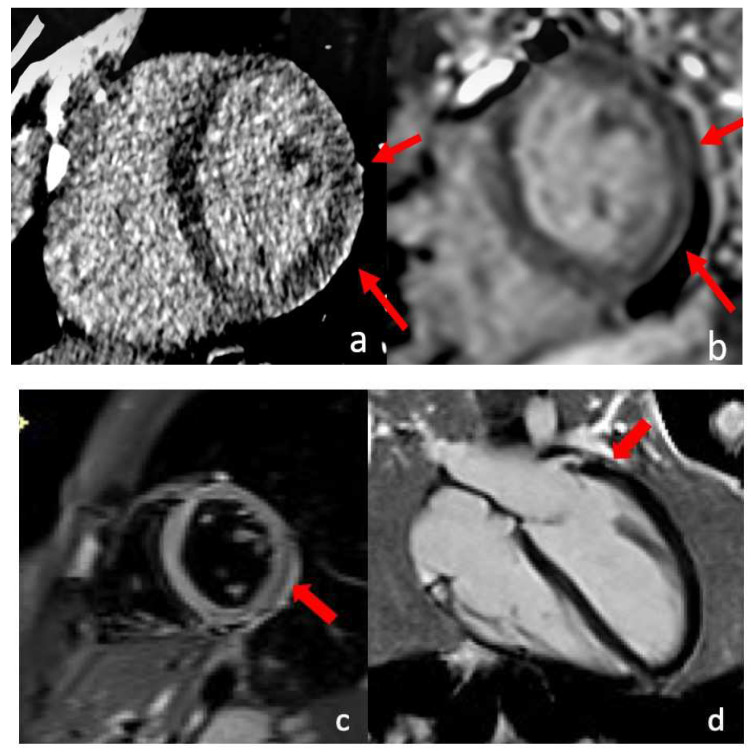
Case 6. Myocarditis: Coronary evaluation at CCTA was followed by a late iodine enhancement (LIE) acquisition (**a**), showing the presence of a myocardial scar (red arrow) in the inferolateral wall of the basal left ventricle. The same findings (red arrow) was confirmed upon the subsequent CMRI on the LGE acquisition (**b**). CMRI was able to demonstrate the presence of acute damage (edema) on STIR sequences (red arrow) (**c**) and acute hyperemia (red arrow) on early gadolinium enhancement (EGE) acquisition (**d**).

**Figure 11 diagnostics-13-02611-f011:**
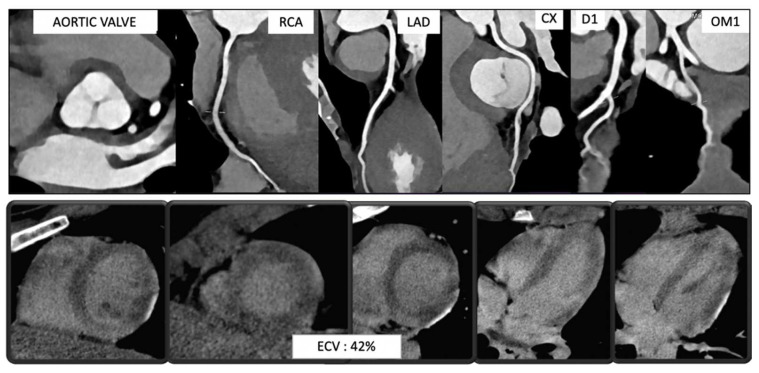
Case 7. Myocarditis: A 35-year-old woman underwent CCTA, using the dual-energy technique (DECT), for thoracic discomfort in the presence of fever. Patient was classified as a low ACS risk. CCTA excluded significant coronary stenosis (**upper** row of figure). A delayed acquisition dual-energy phase was performed after an 8 min contrast injection, using iodine maps to quantify ECV (**lower** row of figure). The ECV calculated was higher than normal, and several areas of late iodine enhancement were seen.

**Figure 12 diagnostics-13-02611-f012:**
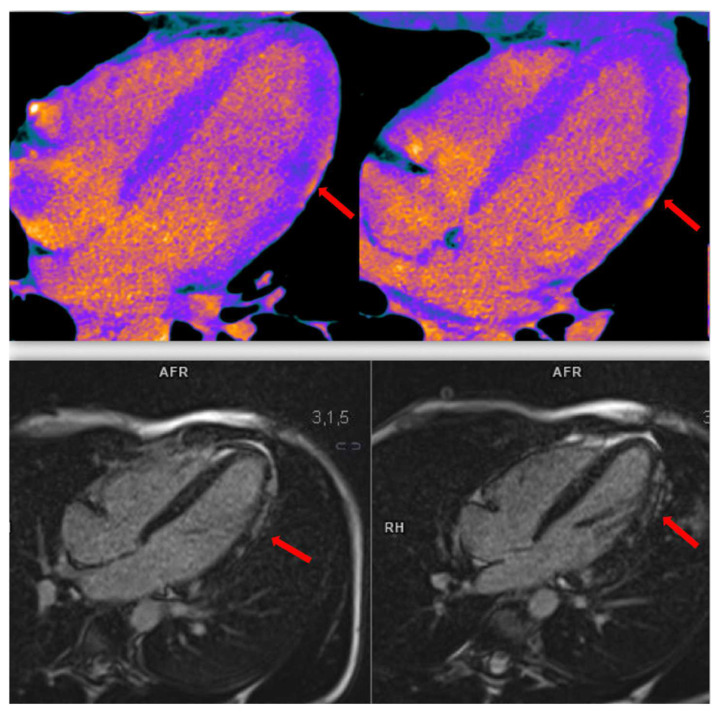
Case 7. Myocarditis: In the upper row, four chambers iodine maps showing hyperenhancement foci in the subepicardial layer along the left ventricle lateral wall (red arrow). In the lower row, same patient underwent a subsequent CE-CMR study showing, in late gadolinium enhancement phase, a perfect overlap of findings compared to DECT.

**Table 1 diagnostics-13-02611-t001:** Main CCTA and CMR findings in women with chest pain, according to clinical scenarios.

	CCTA	CMR
Acute Coronary Syndrome	- Noninvasive significant stenosis (>50%) depiction	- Acute myocardial injury depiction (edema-sensitive sequences) - Post-ischemic fibrosis depiction (LGE) - Reduced blood myocardial supply (stress imaging)
MINOCA	- Noninvasive significant (>50%) stenosis exclusion - CAD depiction	- Acute myocardial injury depiction (edema-sensitive sequences) - Post-ischemic fibrosis depiction (LGE) - Reduced blood myocardial supply presence (stress imaging)
SCAD	- Coronary wall hematoma depiction - Coronary lumen obstruction evaluation - Intimal flap presence	- Acute myocardial injury depiction (edema-sensitive sequences) - Post-ischemic fibrosis depiction (LGE) - Coronary wall hematoma (T1 sequences)
Takotsubo Syndrome	- Noninvasive significant (>50%) stenosis exclusion	- Absence of myocardial structural damage (LGE) - Contractility alteration (cine-RM)
Myocarditis	- Noninvasive significant (>50%) stenosis exclusion - Late iodine enhancement depiction (dual-energy study) - ECV evaluation	- Acute myocardial injury depiction (edema-sensitive sequences) - Post-ischemic fibrosis depiction (LGE) - Native T1 and Native T2 mapping evaluation
Pericarditis	- Pericardial effusion - Pericardial thickening - Late iodine enhancement (LIE) presence - Pericardial constriction signs on cardiovascular structures	- Acute pericardial injury depiction (edema-sensitive sequences) - Post-ischemic fibrosis depiction (LGE) - Pericardial effusion - Signs of pericardial constriction - Functional impairment (cine-RM)

## Data Availability

Data are available upon request.

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
