# Peer review of "Advanced Cardiac Imaging and Women’s Chest Pain: A Question of Gender"

_diagnostics, 2023, doi:10.3390/diagnostics13152611_

Round 1
Reviewer 1 Report
Authors conducted a review of using advanced imaging modalities in the diagnosis of women with chest pain. it is a good review article having clinical value and impact. However, after reading through it, I have the feeling that it is insufficient in terms of contents on the imaging modalities as authors only focused on cardiac CT and MRI (in parts, echocardiography was mentioned). This is apparently insufficient given the increasing use of so many imaging modalities. Thus I would suggest that authors enhance their review by adding the following contents:
1. Dual energy CT, photon counting CT which represents the latest CT technology.
2. Fused imaging modality: SPECT/CT, PET/CT. and PET/MRI.
3. Most of your paragraphs are quite lengthy so suggest breaking into shorter ones.
4. For an imaging review article, please add more images corresponding to each modality as this will enhance the value of your review article.
5. Pay attention to some spacing issues as i noticed in some parts, there is no space between the sentences.
6. Check the referencing format according to Diagnostics.
Overall scientific writing is fine and acceptable. However, many paragraphs are too lengthy hard to follow, thus I would suggest that authors break down these long paragraphs into shorter ones so that it is easy for readers to follow your contents.
Author Response
Dear colleague, we thank you for reviewing our article and making valuable suggestions. According to your kind requests:
Thus I would suggest that authors enhance their review by adding the following contents:
- Point 1:Dual energy CT, photon counting CT which represents the latest CT technology.
Response 1: We have enhanced the review by adding a paragraph about Dual energy CT, photon counting CT which represents the latest CT technology as requested.
- Point 2:Fused imaging modality: SPECT/CT, PET/CT. and PET/MRI.
Response 2: We have enhanced the review by adding a paragraph about fusion imaging modality: SPECT/CT, PET/CT. and PET/MRI as requested.
- Point 3:Most of your paragraphs are quite lengthy so suggest breaking into shorter ones.
Response 3: We simplified as requested the composition of paragraph according to your kind request.
- Point 4:For an imaging review article, please add more images corresponding to each modality as this will enhance the value of your review article.
Response 4: We added three more clinical cases indicating the use of advanced imaging techniques on the topic of the article as requested.
- Point 5:Pay attention to some spacing issues as i noticed in some parts, there is no space between the sentences.
Response 5: We fixed spacing issues.
- Point 6:Check the referencing format according to Diagnostics.
Response 6: And finally checked referencing format.
Kind regards
The authors
Reviewer 2 Report
The authors present the imaging characteristics in women with acute chest pain, in different clinical scenarios.
First of all the English used in the manuscript is sometimes confusing, hard to be followed, a thorough revision of the text from this point of view would be welcomed.
The manuscript otherwise is well constructed, full with the most relevant data regarding the topic. However, the presence of a table with the main imaging characteristics (CCTA, MRI) found in women in the clinical scenarios presented, would be mandatory for the busy clinician reader.
Has to be improved substantially, mainly by using more readable sentences.
Author Response
Dear colleague, we thank you for reviewing our article and making valuable suggestions. According to your kind requests:
Point 1: First of all the English used in the manuscript is sometimes confusing, hard to be followed, a thorough revision of the text from this point of view would be welcomed.
Response 1: Revision of the text was made in order to enhance English quality and to simplify paragraphs nomenclature according to your kind request.
Point 2: The manuscript otherwise is well constructed, full with the most relevant data regarding the topic. However, the presence of a table with the main imaging characteristics (CCTA, MRI) found in women in the clinical scenarios presented, would be mandatory for the busy clinician reader.
Response 2: We added a table with the main imaging characteristics found in women in the clinical scenarios of acute chest pain as requested.
Kind regards
The authors
Round 2
Reviewer 1 Report
Thank you for your effort in revising the manuscript and addressing my comments. The revised version is much improved.
Overall language is acceptable. I noticed in several places, authors used the word "Anyway", which is not common. I will leave it for the proof reading/language editing stage to fix it.
Reviewer 2 Report
Ther manuscript was improved significantly. Please check for minor ortographic errors (i.e., ematoma vs. hematoma in the table)
ok, minor ortographic errors